# An Improved GC-MS Method for Malondialdehyde (MDA) Detection: Avoiding the Effects of Nitrite in Foods

**DOI:** 10.3390/foods11091176

**Published:** 2022-04-19

**Authors:** Wenjie Wang, Zhiwen Zhang, Xiaoying Liu, Xiaoji Cao, Lianzhu Wang, Yuting Ding, Xuxia Zhou

**Affiliations:** 1College of Food Science and Technology, Zhejiang University of Technology, Hangzhou 310014, China; wangw0901@zjut.edu.cn (W.W.); chihwencheung@163.com (Z.Z.); liuxiaoying04@126.com (X.L.); dingyt@zjut.edu.cn (Y.D.); 2Key Laboratory of Marine Fishery Resources Exploitment & Utilization of Zhejiang Province, Hangzhou 310014, China; 3National R&D Branch Center for Pelagic Aquatic Products Processing (Hangzhou), Hangzhou 310014, China; 4Collaborative Innovation Center of Seafood Deep Processing, Dalian Polytechnic University, Dalian 116034, China; 5Research Center of Analysis and Measurement, Zhejiang University of Technology, Hangzhou 310014, China; xiaojicao@zjut.edu.cn; 6Yellow Sea Fisheries Research Institute, Chinese Academy of Fishery Sciences, Qingdao 266071, China; lianzhu_wang@aliyun.com

**Keywords:** malondialdehyde, lipid peroxidation, pickled products, nitrite, GC-MS

## Abstract

Malondialdehyde (MDA) is one of the representative end products under lipid peroxidation, indicating the degree of lipid oxidation in foods. However, compounds in pickled products, especially the nitrite in salted lean pork can react with MDA under the acidic condition, leads to the loss of MDA and an underestimation on lipid oxidation through the conventional assay. In this study, the quantification for MDA in the sample containing sodium nitrite were found lacking accuracy by the thiobarbituric acid (TBA) assay and chromatography assay based on alkaline hydrolysis as the reaction between them were difficult to be completely inhibited. Among other trials, the improvement GC-MS analysis utilizing deuterium substituted MDA (MDA-d_2_) as an internal standard and applying perfluorophenylhydrazine (PFPH) as a derivative reagent can reduce the deviations from the presence of nitrite in the salted lean pork meat and the recovery is between 93.9% and 98.4% and coefficient of variation for the intermediate precision is between 1.1 and 3.5% using the method. The advanced gas chromatograph mass spectrometer (GC-MS) assay also has a very low detection limit (0.25 ng/mL) with both hydrolysis types.

## 1. Introduction

Lipid peroxidation occurs in meat foods and produces various oxidation products, including aldehydes, ketones and short chain fatty acids (SCFAs). In the group of aldehydes, MDA is a relatively dominate and stable form. It was commonly used to indicate the degree of lipid oxidation in foods [1]. There are two forms of MDA detected in foods, free malondialdehyde (FrMDA) and protein-bound malondialdehyde (PrMDA) which in the unstable Schiff base conjugates form. The later one is dominated in meat matrix, the linkages between MDA and amino acids are easily able to obtain hydrolyzed by either acid or alkali treatments. So far, the determination of MDA in food-related studies are mainly based on detecting total amount of both MDA forms.

A thiobarbituric acid (TBA) assay is one of the most commonly used methods for determining the MDA content [2]. By adding a high concentration of trichloroacetic acid (TCA) to hydrolyze PrMDA, the free MDA could be released through the treatment of TBA addition at ~90 °C high temperature for 90 min under an acidic environment. The absorbance of the TBA-MDA complex is commonly measured at 532 nm. However, TBA does not specifically react with MDA. Those interfered compounds are also known as thiobarbituric acid reactive substances (TBARs), have similar spectra properties. They are phenolics [3], water-soluble proteins and peptides [4], other carbonyl compounds from oxidized lipids [5], sucrose and some pigments [6], leading to an over-estimation on MDA quantification. Antonios et al. (2012) reported that the MDA reads from the conventional spectrophotometric TBA assay on dried nuts, sausages, cooked fish and cheese are lack of accuracy due to the color interference from other lipid oxidation products [7]. Recent application of chromatographic techniques significantly improves the precision of TBA-MDA detection due to its elevated separation performance, which can best eliminate the influence from TBARs. Progresses have been achieved by the high-performance liquid chromatography (HPLC) assays. For instance, the MDA-2,4,dinitrophenylhydrazine (DNPH) adduct was able to be detected by the high-performance liquid chromatography with ultraviolet (HPLC-UV) [8,9], the MDA-TBA adduct could be detected by HPLC-fluorescence detector (FLD) with a higher sensitivity [10,11]. However, TBA cannot be used as a derivative reagent in GC analysis on MDA due to the low volatility of TBA-MDA. Instead, derivative reagents, including *N*-methyl hydrazine (NMH) [12], phenylhydrazine (PH) [13], 2,4,6-trichlorophenylhydrazine (TCPH) [14], PFPH [15,16], *O*-(2,3,4,5,6-pentafluorobenzyl) hydroxylamine (PFBHA) [17] and pentafluorobenzyl bromide (PFB-Br) [4,18] can be used. The formation reactions of these MDA derivatives were shown in Figure 1. Moreover, the sensitivity of the GC method is much higher than the colorimetric TBA assay [19,20].

Moreover, MDA quantification of meat products might easily read wrong due to the presence of nitrite. Cured, fermented, brined meats or other foods contain high levels of nitrite, derived from nitrifying bacteria or intentional addition in commercial to protect the food from color changes and oxidation. Furthermore, nitrite can easily react with MDA under the acidic conditions, and most detection methods use acid treatment for hydrolysis and derivatization, which leads to an underestimation on the MDA content. Some studies, thus, attempted to add sulfonamide to remove the presence of nitrite. But sulfonamide was found also reacting with MDA and could produce a 1-amino-3-iminopropylene derivative [21]. From Jung et al., it was confirmed that the method of detecting MDA-DNPH with HPLC-UV after alkaline hydrolysis as well as the traditional TBA assay cannot accurately quantify MDA in cured meat products, which indicates that the conventional acid hydrolysis or derivatization conditions would lead to a significant reaction between MDA and nitrite, so they proposed to analyze MDA directly by HPLC-UV without derivatization after alkaline to avoid the interference of nitrite [22]. However, the direct measurement on MDA with HPLC was still not common because of the poor stability and weak signal response from MDA [23]. In addition, many studies reported a significant difference between the MDA values measured after the acid hydrolysis and the alkaline hydrolysis [10], so it is not conducive for scholars to compare the results obtained by the method based on alkaline hydrolysis with those obtained by the traditional method.

Therefore, this research paper aims to develop an advanced detection method based on derivatization that is not limited by hydrolysis type method. To achieve this goal, three improved GC-MS methods relying on external standards and four on internal standards were compared with each other and validated afterward. The deuterated internal standard method which showed a more accurate quantification on MDA in the meat product model is optimized and compared with the previously proposed HPLC method.

## 2. Materials and Methods

### 2.1. Reagents

All chemicals and reagents used in the study were of analytical or HPLC grade with the highest purity. 1,1,3,3-Tetraethoxypropane (TEP), 2,6-di-tert-butyl-4-methylphenol (BHT), sodium nitrite, trichloroacetic acid (TCA, CCl_3_COOH), disodium hydrogen phosphate dodecahydrate, citric acid and phenylhydrazine including PH, DNPH and TCPH, n-hexane and acetonitrile (HPLC grade) were purchased from Aladdin (Shanghai, China). 1,1,3,3, tetraethoxypropane-d_2_ (TEP-d_2_) and 2-methyl-3-ethoxypropenal, PFPH were purchased from Sigma (St. Louis, MS, USA). TBA was purchased from Sinopharm Chemical Reagent Co., Ltd. (Shanghai, China), and hydrochloric acid (37%) and sodium citrate were purchased from XiLong Science (Shanghai, China).

### 2.2. Standard Solutions

MDA and MDA-d_2_ standard solutions were prepared as the hydrolyzed products from TEP and TEP-d_2_, respectively. Their concentrations were determined by the absorbance at 244 nm (ε = 13,700) under TBA assay. Specifically, the Methyl-malondialdehyde (MeMDA) solution was prepared by reacting 50 μL of 2-methyl-3-ethoxyacrolein (0.048 g) with 50 μL of 7 M NaOH at 40 °C for 3 min, and the final concentration was diluted to 5 μg/mL with water. Buffer solutions of pH 2.0, 3.0, 4.0, 5.0, 6.0 and 7.0 were prepared with 0.1 M citric acid and 0.2 M disodium hydrogen phosphate. All the solutions were stored at 4 °C before use.

### 2.3. MDA-Nitrite Mixed Solutions and Meat Samples

Six groups of the simulated solutions have constant MDA amount mixed with different concentrations of sodium nitrite, were used to validate the detection method in the study. As the maximum allowable amount of sodium nitrite in food is 150 mg/kg and the general content of MDA in food is between 0.1 and 5.0 mg/kg, the MDA concentration in the simulated solutions was fixed at 1 μg/mL. While the sodium nitrite concentrations were set at 0, 0.95, 4.72, 9.45, 18.89 and 47.22 μg/mL, with MDA-to-Nitrite mole concentration ratios of 1:0, 1:1, 1:5,1:10, 1:20 and 1:50, respectively.

Meat samples with MDA and different Nitrite solutions were prepared as follows: Fresh lean pork meats (protein 21.2 g/100 g; fat 6.4 g/100 g; carbohydrate 2.5 g/100 g; sodium 58 mg/100 g; moisture 46.9 g/100 g, the data comes from the nutritional composition table printed on food packaging) from a local supermarket (Hangzhou, China) were minced to strips and kept at room temperature for 24 h to produce naturally-formed MDA. Then, they were divided into four different groups by adding sodium nitrite to reach the final concentrations of 0, 0.015%, 0.003% and 0.0005% (*w/w*). The minced meats were dry-mixed for 2 min, then immediately packed and stored at –30 °C before use.

### 2.4. Hydrolysis Treatments on Tested Samples

According to the requirements from different detection methods, all tested samples were selectively hydrolyzed either by acid or by alkali treatment. For acid hydrolysis, 1 mL of mixed solution or 2 g of meat samples, 100 μL of 2 mg/mL BHT solution, 400 μL of 5 μg/mL internal standard (MDA-d_2_ or MeMDA, if necessary) and 4 mL of 10% (*w/v*) aqueous TCA solution was added to a 30 mL centrifugal tube, and deionized water was added to reach a final volume of 8 mL. The mixture was then homogenized and centrifuged at 10,000 r/min, 5 min prior to filtration. A total of 4 mL of the filtrate was transferred into another centrifugal tube and adjusted to pH 2.0 or 3.0 with NaOH solution. Deionized water was added to produce a final volume of 5 mL. Then, 3 mL of a buffer solution with the same pH was added to obtain an acidic hydrolysate solution. For the alkaline hydrolysis, the mixture was prepared by a similar way to the acid hydrolysis method, while the TCA solution was replaced by 2.0 M NaOH. Plus, the mixture was heated at 60 °C for 60 min to achieve a sufficient hydrolysis on the Schiff base. The pH of the reaction solution was adjusted to 2.0 or 3.0 with an HCl solution, and deionized water was added to reach a final volume of 9 mL. The homogenized mixtures were centrifuged at 10,000 r/min for 5 min, and filtered through filter paper. Alkaline hydrolysate solutions were obtained by mixing 4.5 mL of the filtrate with 3.5 mL of the buffer solution with the same pH.

### 2.5. MDA Detection Assays

#### 2.5.1. Detection of MDA by Conventional TBA Assay

First, 5 mL of acidic or alkaline hydrolysate solution under pH 2.0 was derivatized with 5 mL of 2 mg/mL TBA by heating in a 90 °C water bath for 90 min. Then, the mixture was cooled to room temperature and got sonicated, and the absorbance was measured at 520 nm.

#### 2.5.2. Detection of MDA Directly by HPLC-UV Assay

This procedure was performed according to the method described by Jung et al. (2016) [22]. First, 16 mL of acetonitrile was added to an alkaline hydrolysate solution to extract the MDA, and 1 mL of the extract was filtered through a 0.45 μm filter for the HPLC-UV analysis. The HPLC analysis was performed using a WATERS (Milford, MA, USA) model E2695 HPLC system with a WATERS XBridge-C_18_ column (4.6 mm × 250 mm, 5 μm). Phosphoric acid was used to regulate the pH of flow phase (0.02 M K_2_HPO_4_) to 6.4. The flow rate of the mobile phase was set at 1.0 mL/min, with a 10 μL injection volume. The column temperature was maintained at 25 °C, and the detection wavelength was 254 nm. The retention time of MDA was ~2.9 min.

#### 2.5.3. Detection of MDA-DNPH by HPLC-UV Assay

The detection of the MDA-DNPH adduct was modified according to the method from Ma and Liu (2017) [8]. First, 2 mL solution of acidic or alkaline hydrolysate solution of pH 2.0 was derivatized with 100 μL of a 2.5 μM DNPH solution, heating at 50 °C for 60 min in the dark. Then, 2 mL of n-hexane was added to extract MDA-DNPH. Then 1 mL of the upper n-hexane extract was further dried through the anhydrous sodium sulfate powders and then went through a 0.45 μm filter for HPLC-UV analysis. The HPLC-UV analysis conditions for MDA-DNPH were the same as session 2.5.2, but the mobile phase was replaced by an acetonitrile and water (50:50, *v/v*) mixed solution for the liquid phase separation. The UV/Vis detector was set at 310 nm. The retention time of MDA-DNPH was around 7.3 min.

#### 2.5.4. Detection of MDA-PFPH by GC-MS Assay

The MDA-PFPH detection procedure using GC-MS assay was slightly modified based on Fan’s method (2002) [24]. First, 2 mL of acidic or alkaline hydrolysate solution (pH 3.0) was derivatized with 100 μL of 2.5 μM PFPH at 50 °C for 30 min. Then, 2 mL of n-hexane was added to extract MDA-PFPH. 1 mL of the upper n-hexane extract was dried by going through anhydrous sodium sulfate powder and a 0.45 μm filter prior to GC-MS analysis. The applied gas mass spectrometer was from THERMO with model TRACE 1300/ISQ 7000 (Waltham, MA, USA) equipped with EI sources. A DB-5 AGILENT column (60 m × 0.32 mm, 1.00 μm) (Santa Clara, CA, USA) was used. The oven temperature was increased from 50 °C to 280 °C at A speed of 15 °C/min, maintained at 280 °C for 8 min. The total operating time was up to 23.33 min. Helium (99.999%) with a flow rate of 1.0 mL/min was used as a carrier gas. The mass spectrometer detector (MSD) condition applied as follows: 250 °C capillary direct interface temperature and 70 eV ionization energy. Furthermore, a full-scan mass spectra (*m*/*z* 40–500) was recorded for the analyte identification, and the retention times were 12.31 min for MDA-PFPH and MDA-d_2_-PFPH, and 13.54 min for MeMDA-PFPH. The derivatives were analyzed by GC/MS under the selected ion monitoring (SIM) mode, and ions of *m*/*z* 234, *m*/*z* 236 and *m*/*z* 248 were selected for the MDA-PFPH, MDA-d_2_-PFPH and MeMDA-PFPH analyses, respectively.

### 2.6. Analyte Solution Stability and Method Validation

To compare two deuterated internal standard methods using GC-MS assay for the directly detection of MDA, the HPLC-UV procedure proposed by Jung et al. (2016) was applied, evaluation on the stability of the analyte solution, as well as the detection limit (LOD), quantitative limit (LOQ), recovery and accuracy among three methods were conducted to validate the optimal assay. For the stability analysis, the analyte solution prepared by the method which detecting MDA directly by HPLC-UV was divided into 8 groups and stored at 4 °C. The analyte solution prepared by the deuterated internal standard method of GC-MS was divided into 16 groups, in which 8 groups were stored at room temperature and the others at 4 °C, then stored samples were obtained at intervals of 0 h, 1 h, 4 h, 12 h, 1 d, 2 d, 4 d and 7 d, respectively. The LOD and LOQ of the methods were determined by instrumental readings, diluting the standard solutions to their corresponding with a signal-to-noise ratio of 3 and 10, respectively. For the recovery and accuracy analyses, pork meat with different concentrations of sodium nitrite (0%, 0.0005%, 0.003% and 0.015%) were used as samples. The recovery rate was calculated by the MDA values of the original pork samples and the MDA added pork samples; 0.1 mL of a 10 μg/mL MDA solution was added to 2 g of the pork sample before hydrolysis to prepare standard addition samples. To test the repeatability, the content of MDA in the pork was determined three times a day for two consecutive days.

### 2.7. Statistical Analysis

All data were expressed as the mean ± SE. The results were analyzed by the analysis of variance (ANOVA) and the significant differences among the means were determined by the Tukey-Kramer test. Statistical significance was assumed at *p* < 0.05.

## 3. Results

### 3.1. Comparison among Three External Standard Methods

Most studies quantified MDA in food by established external standard methods, mainly the colorimetric TBA assay. However, external calibration is lack of accuracy since the target MDA react with nitrite during the acid hydrolysis and derivatization processes regarding the nitrite-containing foods. To avoid the reaction between nitrite with MDA, alkali could be used as hydrolysis reagent instead of acid [22]. Meanwhile, highly reactive derivatization reagents can be used to achieve the same reaction competition, but so far there has no relevant research on this subject. Therefore, our group investigated three representative derivatization reagents (TBA, DNPH and PFPH), the reactants were detected by the spectrophotometer (Method A_2_; the subscript represents the hydrolysis methods used in the pretreatment process, where _1_ means acid hydrolysis and _2_ means alkaline hydrolysis), HPLC-UV (Method B_2_) and GC-MS (Method C_2_). As shown in Figure 2A, the detection of MDA derivatives in the sample solutions with the presence of sodium nitrite were significantly lower than that of the control (no sodium nitrite in the samples) (*p* < 0.05), it confirmed that the reaction between MDA and nitrite occurred during the derivation process. Among all three MDA detection methods applying external standards, the content of the MDA derivatives in all solutions containing sodium nitrite measured by method A_2_ (MDA-TBA/UV-Vis) was significantly lower (*p* < 0.05). In method B_2_ (MDA-DNPH/HPLC-UV) and method C_2_ (MDA-PFPH/GC-MS), the signal intensity decreased in the solutions with a molar ratio of sodium nitrite to MDA ≥ 10, the results proved that the high reactive derivatization reagents could completely inhibit the reaction of low level of sodium nitrite with MDA.

According to Kolodziejska et al. (1990) [25], the critical pH facilitating the reaction between nitrite with MDA was around 6 when the mole ratio of nitrite to MDA in solution was 5, therefore, the study proposed that it might be feasible to inhibit the unwanted reaction by adjusting pH of the solution. However, the rise of pH could also lead to a decrease of derivatization efficiency and affect the sensitivity, so method C_2_ using GC-MS as the detection instrument with low detection limit was selected for the optimization. As shown in Figure 2B, the peak area of the MDA derivatives detection signal for the solutions with a molar ratio of sodium nitrite to MDA ≥ 5 was significantly lower than that of the control (pH 3.0). From the result, with the increase of the derivative pH increased, the difference in the peak area between the solution containing sodium nitrite and the control solution was minimized. For example, the peak area of the group with a molar ratio of sodium nitrite to MDA of 5 is around 7.17% of the control solution (no nitrate) when the pH 3.0. But when the pH reached 7.0, this ratio increased to 59.46%, which indicates that the reaction of MDA with nitrite was considerably inhibited at higher pH. However, if the pH increased to 7.0, formed MDA-PFPH might reacted with high levels of sodium nitrite, which resulted in a considerably lower signal than that of the control. Specifically, the peak area of the MDA derivatives under pH 7.0 was only ~4% of that at pH 3.0, illustrating a poor sensitivity of the method.

Based on our results, no matter the conventional spectrophotometric or the improved external methods, MDA could always react with higher levels of sodium nitrite in the sample solution (without the matrix effect) during the derivation process. Therefore, the improved external methods might not be widely applicable, especially for the food samples with high levels of sodium nitrite.

### 3.2. Comparison among Four Internal Standard Methods

The application of the isotopically labeled internal standards is considered as a feasible method to mitigate the effect of interferents on the results [26]. However, other paper suggest that there are exceptions [27]. This approach was not widely used to detect MDA in food considering the cost and availability. Therefore, to verify whether sodium nitrite affects the detection of MDA in samples with the isotopically labeled internal standards, our group verified four different internal standard methods with GC-MS analysis, including the deuterated internal standard MDA-d_2_ (Method D_1_ and Method D_2_) and homologous internal standard MeMDA (Method E_1_ and Method E_2_) (Figure 3), to determine whether it is feasible to use an inexpensive internal standard alternative. To verify the calibration capability of two types of internal standard methods, both acid hydrolysis (subscript 1) and alkaline hydrolysis (subscript 2) were considered. As the premise of an accurate determination on MDA content in foods containing nitrite by the internal standard method is that the ratio of MDA derivatives (MDA-PFPH) to internal standard derivatives (MDA-d_2_-PFPH or MeMDA-PFPH) in nitrite-containing samples should be almost the same with that of the control.

Based on our findings, compared with the control, there was no significant difference between the peak area ratio of the MDA-d_2_-PFPH adduct and MDA-PFPH adduct after both acid and alkaline hydrolysis and derivatization, which also indicated that MDA-d_2_ and MDA had the similar reactivity with sodium nitrite (D_1_ and D_2_). Moreover, the trend is consistent in the detected range of this study (a molar ratio of 0 to 50 NaNO_2_-to-MDA).

In terms of the methods using MeMDA as internal standard (Method E), the peak area ratio of the MDA-PFPH adduct to the MeMDA-PFPH adduct in the hydrolysate solution of nitrite-containing sample showed totally different results as that of the control. Specifically, the peak area ratio of the MDA-PFPH to MeMDA-PFPH adducts increased under alkaline hydrolysate (E_2_) while decreased in the acid hydrolysate (E_1_) when the samples possessing the sodium nitrite. The difference between two hydrolysis indicated that the reaction between MDA and nitrite is stronger than the reaction between MeMDA and nitrite in an acidic environment, while MeMDA and sodium nitrite could dominate the reaction in a more alkaline environment. This is attributed to the strong electron-withdrawing ability of two carbonyl groups from MDA to generate alkaline MDA ions, which are easily combined with nitrite, similarly MeMDA reacts with nitrite on the same principle. However, MeMDA could be more easily converted to the enol form under acidic conditions, and then forms the more stable vinyl propanedione, resulting in a lower yield of MeMDA reacting with nitrite.

Above all, the method applying MDA-d_2_ as the internal standard in GC-MS analysis with the derivatization of PFPH can effectively avoid the interference of sodium nitrite in the samples.

### 3.3. Optimization of the Deuterated Internal Standard Method

Even if the deuterium internal standard method can correct the deviation of the results, the reaction between nitrite and MDA still affects the sensitivity of the method, especially for samples with high levels of nitrite. Therefore, further optimization on the derivatization method was conducted to improve the sensitivity of the method. There are several types of derivatizing reagents react with the MDA, including PH, TCPH, PFPH and PFBHA, easily form weak polar products [28]. It was noticed that it was difficult to separate the analyte and deuterated internal standards by chromatography. Unless there are ions for differential quantification between the MDA derivatives and MDA-d_2_ derivatives, it is hard to keep accuracy especially when they are in mixed form. However, there is not much mass spectrum information about the derivatives of MDA and MDA-d_2_, thus the study provided, totally, eight derivatives according to the GC-MS-EI profiles. MDA-TCPH derivatives were shown as an example. As shown in Figure 4, the characteristic ions of MDA-TCPH are at *m*/*z* 176, 246 and 248, while the corresponding characteristic ions of MDA-d_2_-TCPH are at *m*/*z* 178, *m*/*z* 248 and *m*/*z* 250. Since the relative abundances of *m*/*z* 178 and *m*/*z* 250 in MDA-TCPH are greater than 10% compared to MDA-d_2_-TCPH, so there is no suitable characteristic ion for accurate quantification of MDA. By contrast, the other derivatives did not produce similar ion fragments for interference detection with the EI source. The MDA-PFBHA and MDA-d_2_-PFBHA adducts were quantified by separately monitoring characteristic ions (*m*/*z* 70 and *m*/*z* 72 or *m*/*z* 250 and *m*/*z* 252). Plus, molecular ion peaks of reaction products of MDA or MDA-d_2_ with PFPH or PH were scanned for quantitative analysis (data was not shown).

The reaction conditions of the four MDA derivatives were separately optimized based on experiments. The optimum reaction conditions were as follows: PH: pH 3.0, 25 °C, 30 min; TCPH: pH 3.0, 50 °C, 60 min; PFPH: pH 3.0, 50 °C, 30 min; PFBHA: pH 5.0, 25 °C, 30 min. The derivative products were extracted by n-hexane and then detected by GC-MS in three scanning modes, including full scan (*m*/*z* 40–500), monitoring base peak ions in the SIM mode, and monitoring ions for quantification in the SIM mode, their signal-to-noise ratios (SNR) were compared. The base peak ions scanned in the SIM mode were *m*/*z* 144 for MDA-PH, *m*/*z* 246 for MDA-TCPH, *m*/*z* 234 for MDA-PFPH and *m*/*z* 181 for MDA-PFBHA. The ions used for quantification scanned in the SIM mode were *m*/*z* 144 for MDA-PH, *m*/*z* 234 for MDA-PFPH and *m*/*z* 250 for MDA-PFBHA. Although the first two scanning modes are usually not selected for detecting MDA in samples containing isotope internal standards, we still compared derivative reagents under these two modes for reference. In addition, since MDA-PFBHA has three isomers and the peaks of isomers 2 and 3 partially overlap. Therefore, they can be quantified either by the sum of the peak areas of the three isomers (MDA-PFBHA) or by the peak area of isomer 1 (MDA-PFBHA*). Note that some scholars questioned that the S/N ratio of isomer 1 to the total product might not be constant in all cases, probably inappropriate in quantifying the MDA content [29].

In Figure 5, the order of SNR of all four derivatives under full scan mode from the largest to smallest was as follows: MDA-PH, MDA-PFPH, MDA-PFBHA, MDA-TCPH and MDA-FPBHA*, while it was monitored that MDA-PFPH had the largest SNR under the base peak ions in the SIM mode due to the lower baseline noise of *m*/*z* 234, and the SNR of MDA-PH is higher than MDA-PFBHA and MDA-TCPH. But it is noticed that PH is more toxic and less stable than PFPH [30]. Therefore, the method that MDA transformed into MDA-PFPH was selected as the best assays with the highest sensitivity.

### 3.4. Comparison among Detection Methods Applied in Food Samples

To verify the accuracy and reliability of methods with MDA-d_2_ as internal standard during the GC-MS assay, D_1_ and D_2_, the MDA contents in pork samples with or without (control group) the addition of nitrite were detected individually. The MDA content in the three sodium nitrite addition groups covered the highest anthropogenic addition range of sodium nitrite allowed by the Chinese national food standards, the highest residual of nitrite allowed left in typical cured products and the common content of sodium nitrite in cured products. In addition, we compared methods D_1_ and D_2_ with the other typical MDA detection methods, including the traditional spectrophotometric TBA assay (method A_1_), HPLC analysis of the MDA-DNPH adduct from samples hydrolyzed by TCA (method B_1_) and HPLC analysis of MDA from samples hydrolyzed by NaOH (method F_2_). Methods A_1_ and B_1_ are the most commonly-used methods to detect MDA, and method F_2_ is a method previously proposed by Jung [22] which proved it could accurately evaluate MDA in pickled products.

The results in Table 1 show that there were no significant differences in the MDA content between the pork samples with and without sodium nitrite, as determined by methods D_1_, D_2_ and F_2_. However, the two most common MDA detection methods (i.e., methods A_1_ and B_1_) would underestimate the MDA content in pork samples when containing sodium nitrite. Moreover, it is clear that each methods have their own values for the same sample (control). Notably, the MDA contents measured by the methods with the acid hydrolysis pretreatment were similar and had an error of 7.87%, while that measured by method F_2_ was approximately 1.49 times larger than that measured by method D_2_. Moreover, the MDA contents in the pork samples without sodium nitrite exposed to alkaline hydrolysis pretreatment were 2–3 times larger than those receiving acid hydrolysis pretreatment. This result is contrary to the finding of Jung [22]. It might be due to the reason that MDA was extracted by acetonitrile after alkaline hydrolysis of the pork sample, while acetonitrile extraction was not used when establishing the external standards in his study. In general, the hypothetical reasons for the difference in the results determined by all methods that used pretreatment with different hydrolysis reagents may be due to: (1) there was a difference in the amount of free MDA released by the two hydrolysis methods [31]; (2) acids, alkalis and/or heating promoted the formation of MDA accelerated by the produced hydrogen peroxide during hydrolysis [10], and the amount of MDA added intentionally or released by different hydrolysis methods is commonly different.

### 3.5. Analyte Stability and Method Validation

We further compared several methods to accurately quantify MDA in cured products. For analyte stability, acetonitrile was used to extract MDA to avoid contamination of the chromatographic columns from proteins in method F_2_, and the transfer of water to the acetonitrile layer was inevitable during the acetonitrile extraction. Our group confirmed that the acetonitrile would contain water with strong alkalinity after the extraction of MDA, and the content of MDA in the acetonitrile layer keep decreasing after 24 h and remained only at 86% of the initial content after 7 days. The decrease of MDA might be related with that the high reactivity of MDA in water phase made it easier to obtain self-polymerized [32] or react with other substances, the low stability in alkaline solution might be another reason leading to its decomposition [33]. Supportingly, other study reported that when the MDA extracted from alkaline hydrolysate by n-butanol, it could only remain stable for 3 days even at −20 °C [34]. Therefore, the analytes prepared by method F_2_ must be evaluated within 24 h in addition to store at an ultralow temperature. However, the concentrations of the analytes prepared by method D_1_ or method D_2_ did not considerably change after 7 days of storage at 4 °C. Thus, it was proved that the analyte solutions prepared by methods D_1_ and D_2_ were more stable than those prepared by method F_2_ (data was not shown).

The LOD and LOQ for MDA determination under F_2_, D_1_ and D_2_ were estimated. In details, F_2_ were 1.5 and 4.6 μg/mL for method F_2_ and 0.25 and 0.62 ng/mL for methods D_1_ and D_2_, respectively (data was not shown), indicating that the methods D_1_ and D_2_ were a thousand times more sensitive than method F_2_. In Table 2, all three methods showed satisfactory results in terms of recovery, ranged between 90.7 and 101.3%. For precision, it was shown that the CVs of the intra-day and inter-days analyses of MDA content in the pork samples determined by method F_2_ were higher than those determined by methods D_1_ and D_2_. Therefore, it could conclude that method D was the most optimal assay on MDA evaluation in meat samples.

## 4. Conclusions

This study showed that the conventional TBA assay (Method A_1_) and HPLC-UV (DNPH derivatives) method (Method B) may lead to the underestimation of MDA results in nitrite-containing solution samples. But both MDA/HPLC-UV/NaOH (Method F_2_) method and MDA-PFPH/GC-MS method with MDA-d_2_ as an internal standard (Method D) were not able to be affected by the presence of nitrite in the samples, and the analyte solution from Method F exhibited a better sensitivity and stability as MDA was transformed into PFPH derivatives which well detected by the GC-MS analysis.

Since there are no studies that compare the advantages and disadvantages of both pretreatment methods for hydrolyzing MDA which in the Schiff base form in food samples based on our best knowledge. This study provided the findings that the measured content of MDA in samples hydrolyzed by NaOH is significantly higher than that that by TCA hydrolysis. Conventional acid hydrolysis was proved not viable for the hydrolyzing the Schiff base form of MDA during the pretreatment prior HPLC-UV detection. Detection of MDA by GC-MS using the deuterated internal standard method with either TCA or NaOH to hydrolyze protein-bound MDA during sample pretreatment were found as the most optimal MDA evaluation assay for meat samples even with the presence of interferent and commercially unavoidable nitrite. This paper could be used as a guideline for detecting MDA in pickle meat, and further improvement could be conducted on other pickle products.

## Figures and Tables

**Figure 1 foods-11-01176-f001:**
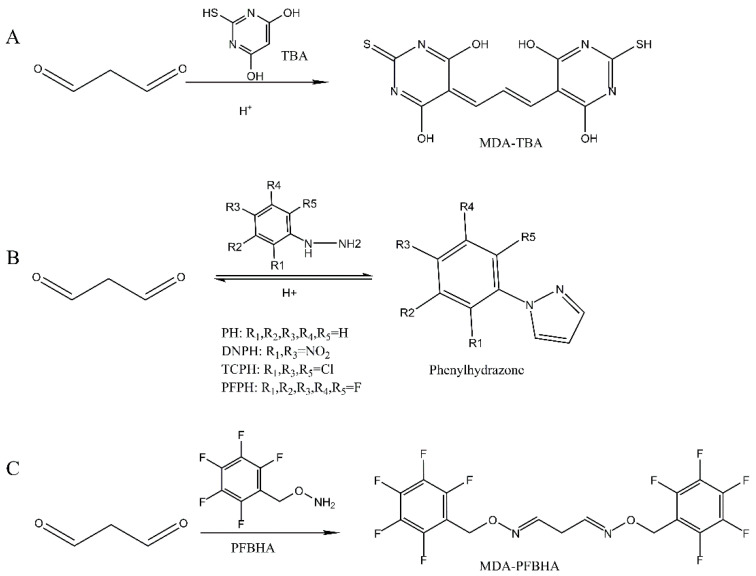
The derivatization reactions between MDA and different derivatization reagents. MDA with TBA (**A**), phenylhydrazine (PH) compounds (**B**) or PFBHA (**C**).

**Figure 2 foods-11-01176-f002:**
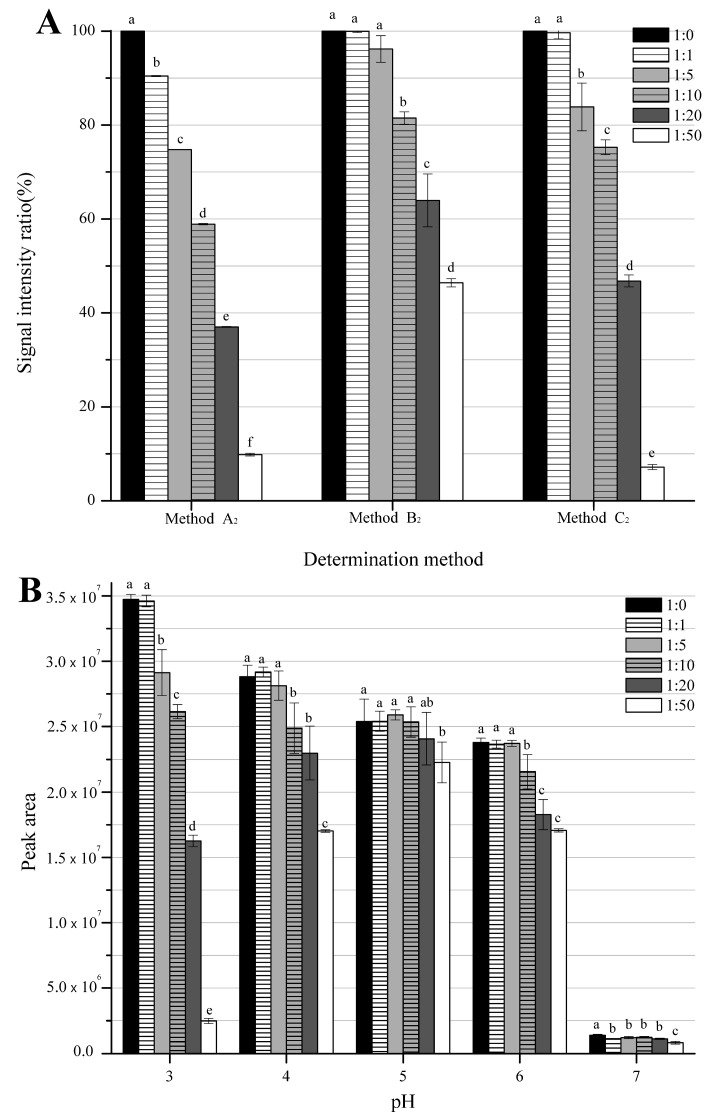
Comparison on MDA detection signal intensity among three external standard methods (Method A_2_: TBA, pH 2.0, 90 °C, 90 min; Method B_2_: DNPH, pH 2.0, 50 °C, 90 min; Method C_2_: PFPH, pH 3.0, 50 °C, 30 min.) (**A**) and peak area through different derivative pH (3, 4, 5, 6 and 7) under MDA-PFPH detection method (**B**) for solution samples containing sodium nitrite with 1:0, 1:1, 1:5, 1:10, 1:20 and 1:50 molar ratio of MDA to sodium nitrite. Noted that samples were hydrolyzed by NaOH in all three methods (subscript as 2), signal intensity ratio indicates the ratio of signal values for each group to the 1:0 group (Control). All reactions under MDA-PFPH detection method (Method C_2_) were carried out at 50 °C for 30 min, noted that pH 3.0 is the most conventional practice (Control). a–f Different letters above the histogram indicates the subgroups showing significant differences in statistics (*p* < 0.05).

**Figure 3 foods-11-01176-f003:**
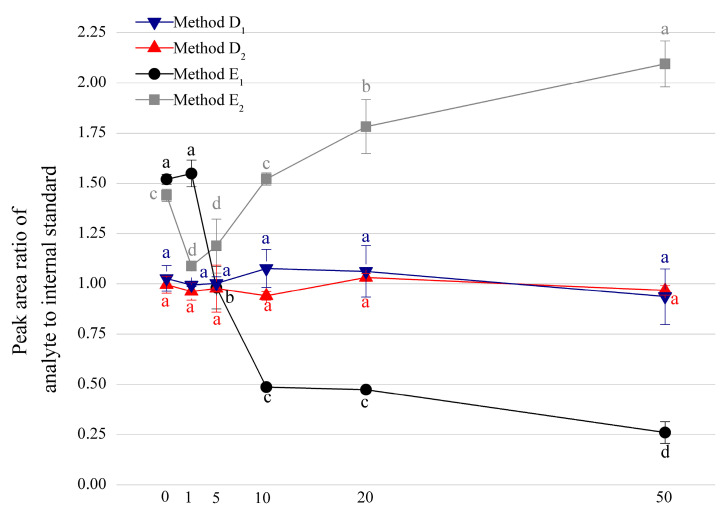
Comparison on MDA detection among four internal standard methods (MeMDA as internal standard with acid and alkaline hydrolysis, E_1_ and E_2_, or MDA-d_2_ as internal standard with acid and alkaline hydrolysis, D_1_ and D_2_) for solution samples containing sodium nitrite with 1:0, 1:1, 1:5, 1:10, 1:20 and 1:50 molar ratio of MDA to sodium nitrite. Noted that the derivatizing agent was PFPH for all methods, and the derivatized products were detected by GC-MS. a–d Different letters above the curve indicates the subgroups showing significant differences in statistics (*p* < 0.05).

**Figure 4 foods-11-01176-f004:**
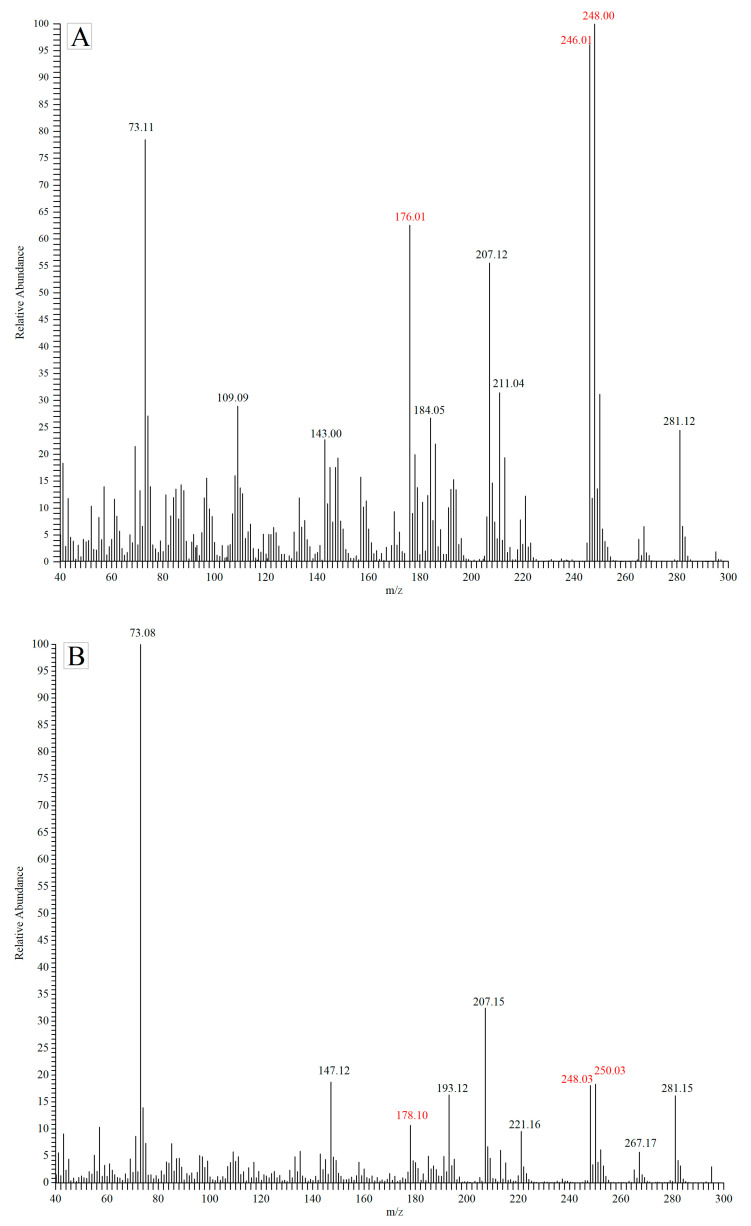
GS-MS-EI profiles of derivatives for two internal standard method (**A**) MDA-TCPH and (**B**) MDA-d_2_-TCPH).

**Figure 5 foods-11-01176-f005:**
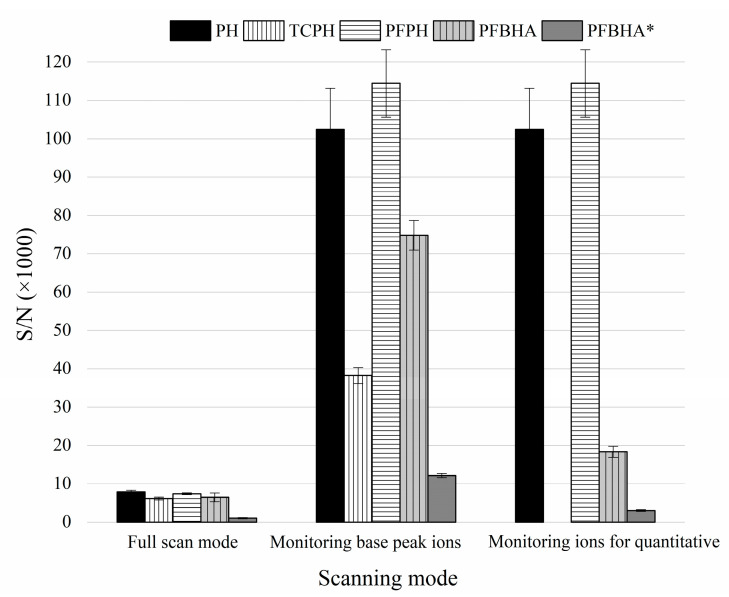
Comparison on the signal-to-noise (S/N) ratios of four MDA derivatives (PH, TCPH, PFPH and PFBHA) under three different EI scanning modes. Noted PFBHA: Sum of S/N of three isomers of the MDA-FPBHA adduct, PFBHA*: S/N of isomer 1 of the MDA-FPBHA adduct.

**Table 1 foods-11-01176-t001:** Quantification of MDA in meat product samples with different addition concentration of sodium nitrite (0, 0.005, 0.03 and 0.015%, *w*/*w* ground pork meat) quantified by five methods (A_1_, B_1_, D_1_, D_2_ and F_2_).

Methods ^a^		Meat Product Models with Different Addition Concentration of Sodium Nitrite			
		Control	0.0005%	0.003%	0.015%
A_1_	MDA-TBA adduct/UV-Vis/TCA	0.704 ± 0.016	0.665 ± 0.009 *	0.660 ± 0.032 *	0.403 ± 0.013 *
B_1_	MDA-DNPH adduct/HPLC-UV/TCA	0.797 ± 0.068	0.522 ± 0.111 *	0.325 ± 0.027 *	0.281 ± 0.008 *
D_1_	MDA-PFPH adduct (MDA-d_2_)/GC-MS/TCA	0.850 ± 0.014	0.854 ± 0.018	0.844 ± 0.026	0.826 ± 0.003
D_2_	MDA-PFPH adduct (MDA-d_2_)/GC-MS/NaOH	1.778 ± 0.019	1.748 ± 0.055	1.756 ± 0.026	1.773 ± 0.073
F_2_	MDA/HPLC-UV/NaOH	2.659 ± 0.061	2.607 ± 0.100	2.731 ± 0.242	2.715 ± 0.179

^a^ Analyte (internal standard)/detection equipment/hydrolysis reagent. All MDA content data were expressed as mg MDA/kg meat product (*n* = 3) with mean ± standard deviation. * Compared with the control group, the MDA detection value under the meat product model differs significantly (*p* < 0.05).

**Table 2 foods-11-01176-t002:** Recovery (%) and precision with intra and inter CV (%) among three MDA detection methods for food samples with different addition concentration of sodium nitrite (0, 0.005, 0.03 and 0.015%, *w*/*w* ground pork meat) (*n* = 3).

Meat Product Models with Different Addition Concentration of Sodium Nitrite	F_2_-MDA/HPLC-UV/NaOH ^a^			D_1_-MDA-PFPH (MDA-d_2_)/GC/TCA			D_2_-MDA-PFPH (MDA-d_2_)/GC/NaOH		
	Recovery (%)	Intra CV (%)	Inter CV (%)	Recovery (%)	Intra CV (%)	Inter CV (%)	Recovery (%)	Intra CV (%)	Inter CV (%)
Control ^b^	90.7	2.3	2.5	96.2	1.7	3.2	98.0	1.0	1.5
0.0005%	101.3	3.8	3.2	98.4	2.1	2.4	97.8	3.1	2.6
0.003%	95.0	8.9	6.5	96.2	3.1	2.4	94.6	1.5	1.5
0.015%	93.4	6.6	4.9	98.3	0.3	1.0	93.9	4.1	3.5

^a^ Analyte (internal standard)/detection equipment/hydrolysis reagent; ^b^ Control: ground pork without nitrite.

## Data Availability

The data that support the findings of this study are available from the corresponding author at reasonable request.

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
