# Peer review of "An Improved GC-MS Method for Malondialdehyde (MDA) Detection: Avoiding the Effects of Nitrite in Foods"

_foods, 2022, doi:10.3390/foods11091176_

Round 1
Reviewer 1 Report
Authors elucidated the analytical method of general lipid oxidation marker, MDA, under the consideration of nitrite presence. Although the study design is clear, following points should be investigated.
- In this study, MDA was detected at ~2.9 min. However, the retention time is mostly same as its holdup time. Therefore, in this condition, the concentartion of MDA is not reliable and must not be discussed. (In Ref. 25, flow rate was 1.2 ml/min, and retention time was 3.1 min.) Authors should improve the HPLC conditions.
- In fig.2 authors described the method C2 was found having higher signal intensity.... however, method B2 looks like higher intensity. Please explain more clearly.
- Discuss the reason why the reaction between MDA and nitrite is stronger than that between MeMDA and nitirite.
Reviewer 2 Report
The document presents the GC-MS method for the malondialdehyde quantification in foods.
I recommend that the authors should define MDA in the title.
The authors should include results in the abstract section.
I recommend that the authors should check the grammar errors (e.i. a underestimation (abstract); it result (introduction)).
The authors should measure the proximal composition of meat samples.
The authors should describe the statistical analysis in the materials and methods section.
The authors should reinforce the discussion section with the latest references.
The references are very old. I recommend that the authors should check and cite the most current references about this topic.
Round 2
Reviewer 1 Report
>But the results of MDA determination in this study are still reliable, referring to Table 2, the recovery of this method for the samples in this study is between 90.7% and 101.3%.
No, recovery also must be calculated by reliable method. Did the authors confirm the MDA separation from a solvent peak and matrix compounds? After these consideration, recovery and MDA content should be discussed.